# Late Local Recurrence after Neoadjuvant Therapy and Radical Resection for Locally Advanced Rectal Cancer

**DOI:** 10.3390/cancers16020448

**Published:** 2024-01-20

**Authors:** Adrian Salega, Marina Münch, Philipp Renner, Klaus-Peter Thon, Wolfgang Steurer, Dina Mönch, Jana Koch, Annika Maaß, Hans Jürgen Schlitt, Marc-Hendrik Dahlke, Tobias Leibold

**Affiliations:** 1Robert Bosch Centre for Tumour Diseases (RBCT), Department of General and Visceral Surgery, Robert-Bosch-Hospital, 70376 Stuttgart, Germany; adrian.salega@rbk.de (A.S.); marina.muench@rbk.de (M.M.); philipp.renner@rbk.de (P.R.); klaus-peter.thon@gmx.de (K.-P.T.); marc.dahlke@rbk.de (M.-H.D.); 2Department of Surgery, Klinikverbund Südwest, Krankenhaus Leonberg, 71229 Leonberg, Germany; w.steurer@klinikverbund-suedwest.de; 3Dr. Margarete Fischer-Bosch Institute of Clinical Pharmacology, 70376 Stuttgart, Germany; dina.moench@ikp-stuttgart.de (D.M.); jana.koch@ikp-stuttgart.de (J.K.); annika.maass@ikp-stuttgart.de (A.M.); 4University of Tübingen, 72074 Tübingen, Germany; 5Department of Surgery, University Medical Center Regensburg, 93053 Regensburg, Germany; hans.schlitt@ukr.de

**Keywords:** rectal cancer, neoadjuvant, radiochemotherapy, outcome, follow-up, surveillance, local recurrence

## Abstract

**Simple Summary:**

Neoadjuvant radiochemotherapy (RCT) and total neoadjuvant therapy (TNT) have significantly reduced local recurrence rates in rectal cancer. However, late local recurrences remain a distinct issue. Most recurrences occur within two years; general follow-up programs typically last for 5 years. We found four patients with late local recurrences > 8 years after surgery. This study highlights that neoadjuvant therapy delays local recurrence.

**Abstract:**

Neoadjuvant radiochemotherapy (RCT) and lately total neoadjuvant therapy (TNT) improved local recurrence rates of rectal cancer significantly compared to total mesorectal excision (TME) alone. Yet the occurrence and impact of late local recurrences after many years appears to be a distinct biological problem. We included *n* = 188 patients with rectal cancer after RCT and radical resection in this study; *n* = 38 of which had recurrent disease (sites: local (8.0%), liver (6.4%), lung (3.7%)). We found that 68% of all recurrences developed within the first two years. Four patients, however, experience recurrence >8 years after surgery. Here, we report and characterize four cases of late local recurrence (10% of patients with recurrent disease), suggesting that neoadjuvant therapy in principle delays local recurrence.

## 1. Introduction

Colorectal cancer remains one of the most prevalent types of cancer, after lung and breast cancer. In 2020, over one million colon cancer cases and over 730,000 rectal cancer cases were reported globally [1]. Rectal cancer accounts for 3.9% of all cancer diagnoses for both sexes and is responsible for 3.2% of cancer-related deaths. The widespread adoption of neoadjuvant radiochemotherapy (RCT) has led to a decrease in local recurrence rates and an improvement in disease-free-survival (DFS). These benefits are significantly amplified when chemotherapy is added to the treatment regimen [2].

Local recurrences occur in 6% of cases following RCT, compared to 13% after postoperative chemoradiotherapy within a five-year post-treatment period [3]. The most common sites for distant recurrences are the lungs, followed by liver metastases and local recurrence. Notably, over 90% of all recurrences manifest within the first three years of monitoring [4]. Despite the decline in local and some distant recurrences, no significant improvement has been observed in overall survival (OS) rates post-neoadjuvant therapy [5].

Postoperative surveillance is crucial in colorectal carcinoma management, particularly as secondary hepatic resection offers another curative opportunity for selected patients [6]. Consequently, a five-year post-treatment surveillance period is recommended in Germany for rectal carcinoma patients following curative surgery [7].

While most recurrences in locally advanced rectal cancer appear within three years post-RCT, there are instances of local recurrences occurring well beyond the standard five-year surveillance period [8].

This study aims to assess the rates of these late local recurrences and evaluate the duration of surveillance. We identified a small subgroup of patients who experienced local recurrences long after routine follow-up had concluded. This cohort may benefit from extended surveillance.

## 2. Methods

### 2.1. Patient Population

We retrospectively analyzed the follow-up data of all patients (*n* = 206), who received RCT followed by resection of the rectal primary at our institution from 2000 to 2017. Patients were identified via a query of our prospective, institutional-based clinical research database. We excluded all patients with R1/2-resections (*n* = 17). Additionally, a patient who required immediate surgery because of stenosis and perforation was excluded. The pretherapeutic imaging and staging consisted of a physical examination, rigid rectoscopy, and computed tomography of the chest/abdomen/pelvis in all patients. Local staging was completed at the beginning of the study using endosonography, and starting in 2005, all patients received pretherapeutic MRI scans of the pelvis. Since the pathologic assessment of the circumferential resection margin (CRM) was implemented into the German Rectal Cancer Guidelines in 2008, and we accrued patients starting in 2000, pathologic CRM data were only available for 46% of patients in our cohort. Only 2.7% of these patients had a positive CRM; therefore, CRM was not included in further analysis. This study was approved by the ethics committee of the University of Tübingen (149/2012BO2) and written informed consent for collecting follow-up data was obtained by all participants. Table 1 shows the clinical characteristics of the patient population.

### 2.2. Neoadjuvant Radiochemotherapy

All patients received external-beam radiation therapy as either 3D-conformable or intensity-modulated radiotherapy (IMRT). The most common regimen consisted of 28 fractions with 1.8 Gy, for a total dose of 50.4 Gy. All patients received preoperative 5-fluorouracil (5-FU)-based chemotherapy. The most common protocol was continuous infusion therapy with 5-FU (225 mg/m^2^/d) for a 6-week continuous cycle. Since 2014, all patients have been recommended adjuvant treatment following neoadjuvant therapy; before 2014, this decision was made on an individual basis. Therefore, 103 patients (50%) in this patient population received adjuvant therapy.

### 2.3. Surgery

All patients underwent radical resection of the rectal primary. For rectal tumors greater than 10 cm from the anal verge, a partial mesorectal excision with transection of the mesorectum 5 cm distal to the distal edge of the rectal cancer was performed. For rectal tumors less than 10 cm from the anal verge, a total mesorectal excision (TME) was performed. In most cases, this was achieved with a low anterior resection (147 cases) with a protective loop ileostomy in 106 cases. Because of low-lying tumors or sphincter involvement, 36 patients (19.5%) were treated with an abdominoperineal resection (APR). Surgery was performed 6–8 weeks (median 6.14) after the completion of RCT.

### 2.4. Local Recurrence

Local recurrence was defined as a recurrent tumor within the pelvis below the promontory. A histological confirmation could be obtained in 93.3% of patients (14 cases). In cases without histological confirmation, a size progression of a pelvic mass was defined as local recurrence.

### 2.5. Statistical Analysis

We used our electronic hospital information system iMedOne (Deutsche Telekom Clinical Solutions GmbH, 50676 Köln, Germany) and GapIT (i-Solutions Health/Mesalvo Mannheim GmbH, 68167 Mannheim, Germany) as the data source. The information was collected and processed anonymously in a secured electronic database. Statistical analysis was performed with Bias for Windows Version 11.06 (Hanns Ackermann, University of Frankfurt, 60590 Frankfurt, Germany) and SPSS Statistics Version 24 (IBM Deutschland GmbH, 71139 Ehningen, Germany). The OS as well as DFS was assessed using Kaplan–Meier survival analysis. Group comparisons in the survival analysis were conducted using the Log-Rank test. Differences between individual subgroups were calculated using the Mann–Whitney test. A *p*-value below 0.05 was considered statistically significant.

## 3. Results

During a median follow-up of 88.5 months, 38 patients (20.2%) experienced recurrences. Local recurrences occurred in 15 patients, equating to a rate of 8.0%. Liver and lung metastases were observed in 12 (6.4%) and 7 (3.7%) patients, respectively. Three patients (1.6%) had multi-site metastases. One patient had a metastatic cervical lymph node as the sole site of recurrence. Figure 1 illustrates the cumulative appearance of recurrences. Table 2 and Table 3 provide an overview of the pathological findings and recurrences. Results of the submitted doctoral thesis have been incorporated [9].

The median duration between surgery and local recurrence was 23 months (range 6–135 months, SD 46.0). No significant difference in initial T-stage was found between patients with and without local recurrence (Mann–Whitney U-Test, *p* = 0.46). The distribution of initial T-stage was identical in the subgroup of patients with local recurrence (69% with cT3, 31% with cT4). This is comparable to the T-stage of the initial staging with 78% of patients with cT3 and 21% cT4 tumors. However, the nodal stage significantly influenced recurrence rates. Patients with positive lymph nodes had a higher rate of recurrence (38.6%) compared to node-negative patients (14.6%, *p* = 0.001). Concerning local recurrence, there was also a higher rate in patients with positive lymph nodes (13.6% local recurrences in node-positive vs. 6.3% in node-negative patients), which was not statistically significant (*p* = 0.16).

Most local recurrences (nine patients) occurred within the first 3 years post-surgery. One patient was diagnosed with recurrent disease 39 months after surgery and another one 49 months after surgery. Four additional patients developed late local recurrences after 8–11 years, accounting for 26.7% of all local recurrences. Patients with local recurrences are depicted in Table 4. 

The time to recurrence varied by site, with distant metastases occurring at a median of 17 months and local recurrences occurring at 23 months post-surgery (*p* = 0.022).

The 5-year OS and DFS rates were 79.2% and 72.6%, respectively. Ten years post-surgery, 58.5% of patients remained alive. The 5-year OS was significantly worse in patients with node-positive tumor stages (ypN+). The 5-year OS in node-negative patients was 84.1% compared to 63.0% for patients with node-positive tumors. After 10 years of follow-up, 64.0% of node-negative patients remained alive compared to only 40.9% of node-positive patients (*p* = 0.001).

The DFS was significantly different in both groups as well, with a 5-year DFS of 79.7% in node-negative patients and 49.5% in node-positive patients (*p* < 0.001). Both OS and DFS in relation to nodal stage are depicted in Figure 2 and Figure 3.

The OS was worse in patients who developed recurrence (local and distant) in comparison to patients with no recurrence with a 5-year OS of 89.6% and 41.3%, respectively. The 10-year OS was 66.0% without recurrence and 26.7% with recurrence (*p* < 0.001). See Figure 4.

OS and DFS also varied based on the tumor’s location within the rectum. Low-lying tumors (<6 cm from the anal verge) had worse OS and DFS compared to tumors located in the middle or upper third of the rectum. Patients with a tumor in the lower third had an OS of 72.1% and a DFS of 67.2%; patients with a tumor in the middle third of the rectum had an OS of 83.8% and a DFS of 77.3% (OS *p* = 0.09, DFS *p* = 0.011). For details, see Figure 5 and Figure 6 and Table 5.

Out of four patients with late local recurrence, only one could be treated curatively with an APR and is still alive after 134 months follow-up. This 73-year-old female patient was initially treated with an AR including TME in September 2008. The preoperative restaging showed an ycT3N2-tumor 5 cm above the anal verge. The histological examination of the specimen revealed an ypT2 with a negative nodal status (0/23 lymph nodes). Due to severe side effects caused by the neoadjuvant RCT, the patient refused adjuvant chemotherapy. Regular follow-up has been completed without evidence of recurrence. However, in January 2017, 99 months after surgery, a routinely performed colonoscopy showed a local recurrence in the anastomotic area. After distant metastases had been excluded via CT scan, the patient underwent an APR. Histologically detected infiltration of the dorsal wall of the vagina made a re-resection necessary. Finally, a R0-resection could be achieved. Later on, a presacral mass was detected in a CT scan in May 2018, 16 months after the APR. A PET-scan could not clearly confirm malignancy, so the interdisciplinary decision was to follow up with the patient. During the following 19 months, the presacral mass was stable.

Another female patient received brachytherapy after a late recurrence was detected and biopsy proven. Initially, the 68-year-old woman was staged T3N0 and underwent AR in August 2004. Histology confirmed a node-negative ypT2 tumor. The patient did not receive adjuvant therapy, and regular follow-up did not reveal any evidence of recurrence. Due to vaginal bleeding, further diagnostic investigation was performed in 2013. A biopsy confirmed a local recurrence at the introitus of the vagina 108 months after AR. A palliative brachytherapy was performed. The patient passed away less than one year later, in August 2014.

A lateral local recurrence with infiltration of the ureter and bladder was found in a male patient after AR in July 2001. The initial histological finding revealed an ypT2ypN1-tumor. Adjuvant therapy was recommended but was refused by the patient. The five years of regular follow-up were unobtrusive. Eventually, the patient underwent further diagnostic investigation in 2012 because of elevated tumor markers (CEA levels). A CT scan confirmed recurrence in the pelvic wall 135 months after surgery (see Figure 7). The tumor was histologically confirmed through transanal biopsy. The therapy consisted of an extended surgery with resection of the urethra and bladder after local radiation and chemotherapy, but an R0-resection could not be achieved. Furthermore, the patient developed lung metastases and received radiotherapy. The patient died 5 years after diagnosis of local recurrence in 2017.

Lastly, one patient developed a presacral mass and received local radiation in combination with chemotherapy, unfortunately without any response. The 68-year-old male initially had a bulky, low-lying T4-tumor with stenosis. He received an end-colostomy prior to RCT and a radical APR afterwards in September 2001. Pathology revealed a node-negative ypT3. The patient refused adjuvant treatment. A CT scan in 2012, which was undertaken because of perineal pain, showed a presacral mass (see Figure 8). CT-guided biopsy confirmed a local recurrence 126 months after APR. The palliative radiation was well tolerated by the patient and was followed by chemotherapy. The patient died in September 2013, 18 months after diagnosis of recurrence.

## 4. Discussion

In this comprehensive study, we examined recurrence patterns in a cohort of *n* = 188 patients diagnosed with locally advanced rectal cancer. All patients underwent neoadjuvant radiochemotherapy, followed by radical resection. The median follow-up period for this cohort was 88 months, providing a robust dataset for long-term outcomes. We identified a subset of patients who developed local recurrence beyond a commonly used follow-up timeframe of five years. This group of patients with “late local recurrences” accounted for approximately one-quarter of all local recurrences (26.7%). Our finding underscores the fact that a select group of patients locally recurs beyond the standard five-year window. This group of patients is biologically interesting and warrants further investigation.

Our survival data are in solid agreement with previously published results from randomized controlled trials. For instance, a significant survival difference based on nodal stage has been well documented and is confirmed by our present study. Pooled data of five major European rectal cancer studies revealed a 5-year OS of 77.9% in pathologically node-negative patients and 53.1% in node-positive patients. These data align with our 5-year OS rates of 84.1% for node-negative patients and 63.0% for node-positive patients. Similarly, the DFS of 79.7% in node-negative patients and 49.5% in node-positive patients in the current trial are consistent with the above data, which reported DFS rates of 65.7% and 40.2%, respectively [10]. We believe that this consistency adds credibility to our findings and suggests that they may well be generalized to a broader population and a true cancer phenomenon. Adding to the above, we also studied the impact of tumor location within the rectum on survival. Our data revealed that patients with tumors located in the middle third of the rectum had superior survival rates (83.8% OS and 77.3% DFS) compared to those with distal tumors (72.1% OS and 67.2% DFS). These findings further corroborate previous data, reporting varying 5-year OS and DFS rates based on tumor location [11].

Lower survival rates in patients with distal rectal cancer can generally be attributed to multiple factors. One significant factor is the surgical complexity associated with these tumors, often necessitating APR, which has been shown to increase the risk of a less favorable oncologic outcome compared to AR [10,12,13].

Our analysis showed a sphincter preservation rate of 81%. This is concordant to randomized trials like the German and Polish Rectal Cancer trials, in which overall sphincter preservation rates of 52% and 69% have been reported [2,3].

Additionally, distal tumors are at a higher risk for lateral lymph node involvement, a factor that has been extensively studied, particularly in Japan [14,15,16]. Although lateral lymph node dissection is a standard procedure in Japan for advanced distal rectal cancer, its ability to influence progression-free survival remains a subject of ongoing research [17]. Nonetheless, some investigators strongly believe that it reduces the rate of local recurrence and should be used for all patients with advanced local disease.

Taking into account the extended observation period of the current study (commencing in the year 2000), we consider it challenging to definitively state if lateral lymph node involvement was responsible for the local recurrences observed, particularly the late ones. Not all our patients with local recurrence underwent initial MRI staging of the pelvis, making it difficult to retrospectively study this aspect. However, it is certainly a possibility, and it is thus of high importance that future studies incorporate more advanced imaging techniques for better preoperative staging, as this is good clinical practice in many regions today.

Albeit the exact mechanism behind late local recurrence remains elusive, several risk factors for late recurrence have been reported, including tumor location, tumor stage at diagnosis, preoperative CEA levels, and lymph node status [18,19]. The most significant prognostic factors for DFS and OS are lymph node status and circumferential resection margin [19,20,21,22]. Additionally, the distribution of positive lymph-nodes has prognostic implications. In this study, shorter survival (OS and DFS) [23] and a high incidence of metastatic disease at the time of surgery [24], were associated with more centrally located positive lymph-nodes. The most important predictors for local recurrence are T-status, nodal status, and residual tumor status (R0 versus R1/R2). Surgery type (TME vs. Conventional) and administration of radiotherapy are also influencers of local recurrence. However, these factors have no effect on DFS or OS [23]. Our cohort only showed a significant difference in nodal status concerning local recurrences. Late local recurrences exhibited no substantial difference in this regard, as three out of four were node-negative. Overall, we assume that all known risk factors favor a relatively early onset of recurrences, and the factors for late onset are not known. Due to the low absolute number, statistically, no factor can be excluded in our cohort.

Taken together, the overall recurrence patterns in our cohort were consistent with the existing literature. Most recurrences manifested within the first three years, primarily in the liver and locally. Hence, the distribution and timing of recurrent disease was similar to that in other studies, with liver metastases appearing first, followed by lung metastases and local recurrences [8]. Interestingly, we observed four cases of recurrent disease after nine years of follow-up (approximately 10% of all recurrences), all of which were local. Although statistical considerations on an epidemiological scale must be taken into further account, this might suggest that an extended follow-up period may be beneficial for early detection of late local recurrences for some patients, increasing the possibility of curative re-resection.

Recent advancements in TNT have shown promising results, particularly in controlling distant metastases [24,25,26]. However, they also make the overall management of rectal cancer even more complicated, since stringent, long-term follow-up becomes crucial, particularly in those patients taking part in watchful waiting trials. Some data from the 5-year follow-up of the RAPIDO trial even suggest that local recurrence rates were higher following TNT as compared to long-course chemoradiotherapy (7.2% in comparison to 3.9% (*p* = 0.049) in patients with R0 resections) [27]. Therefore, the selection of patients who can benefit from TNT must remain an area of ongoing research that has to take into account the problems of late local recurrence discussed in the present work.

In conclusion, our study suggests that neoadjuvant therapy may delay local recurrence in patients with locally advanced rectal cancer in principle. This raises questions about the long-term efficacy of neoadjuvant RCT in reducing local recurrence rates, particularly if lateral nodes are involved. Longer-term follow-up of some patients might be warranted, since good results can be achieved even in the case of local recurrence that makes secondary pelvic exenteration necessary [28,29]. And the very problem of late local recurrence discussed in the present work has become even more pressing with the further introduction of TNT- and watchful-waiting strategies that sometimes trouble the surgeon.

## Figures and Tables

**Figure 1 cancers-16-00448-f001:**
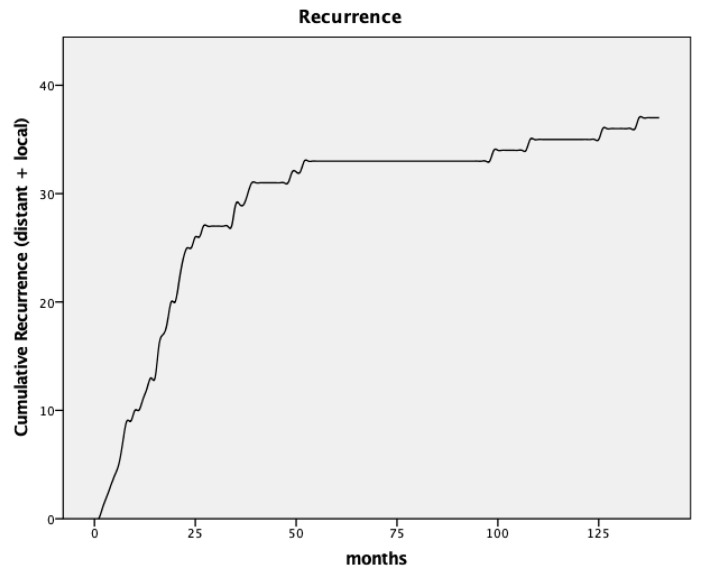
Curve for cumulative recurrence.

**Figure 2 cancers-16-00448-f002:**
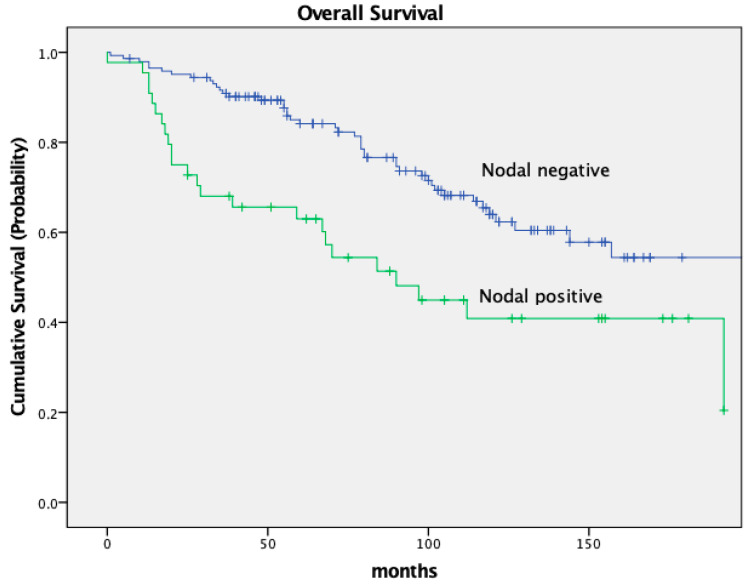
Kaplan–Meier curves for OS, nodal stage.

**Figure 3 cancers-16-00448-f003:**
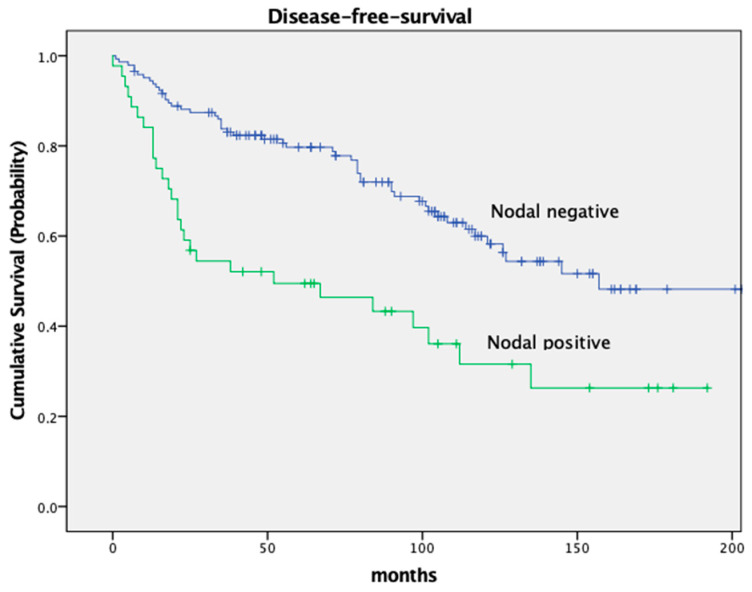
Kaplan–Meier curves for disease-free-survival, nodal stage.

**Figure 4 cancers-16-00448-f004:**
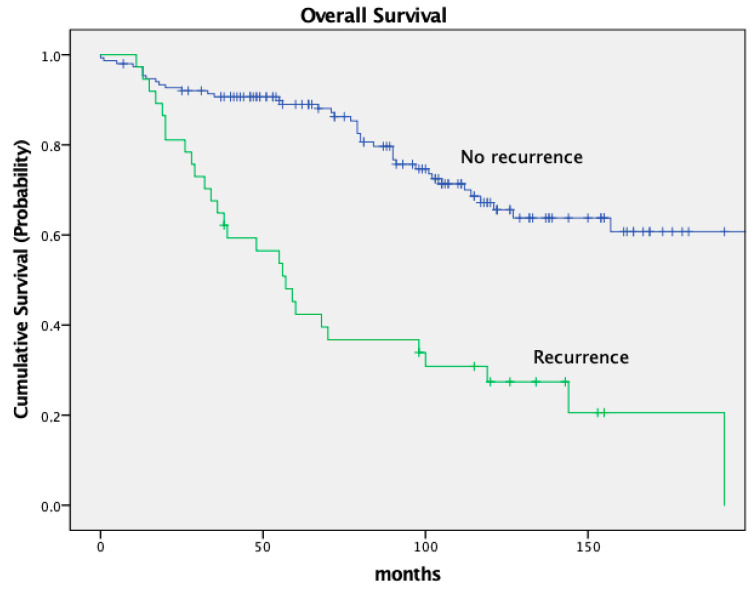
Kaplan–Meier curves for recurrence.

**Figure 5 cancers-16-00448-f005:**
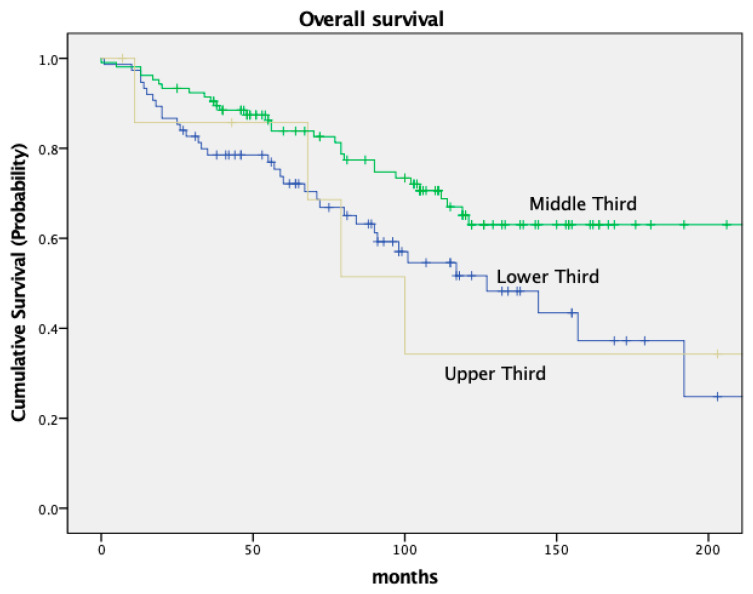
Kaplan–Meier curves for OS, location of tumor.

**Figure 6 cancers-16-00448-f006:**
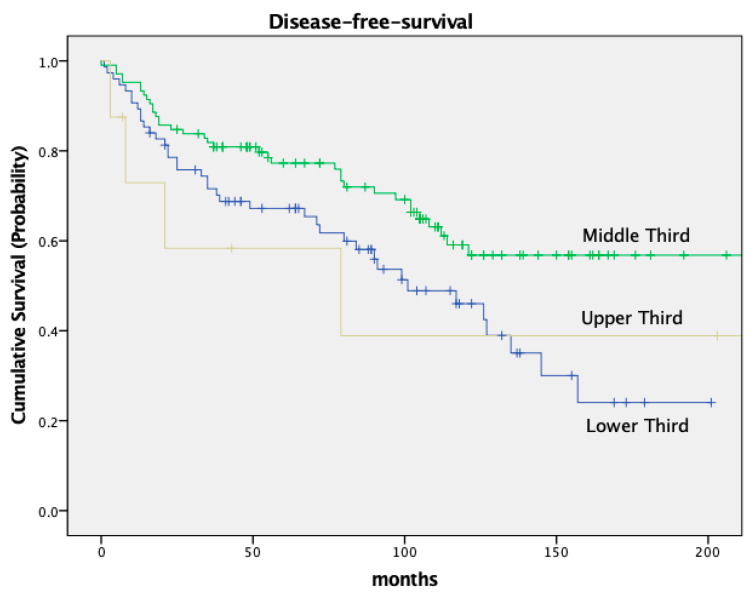
Kaplan–Meier curves for disease-free-survival, location of tumor.

**Figure 7 cancers-16-00448-f007:**
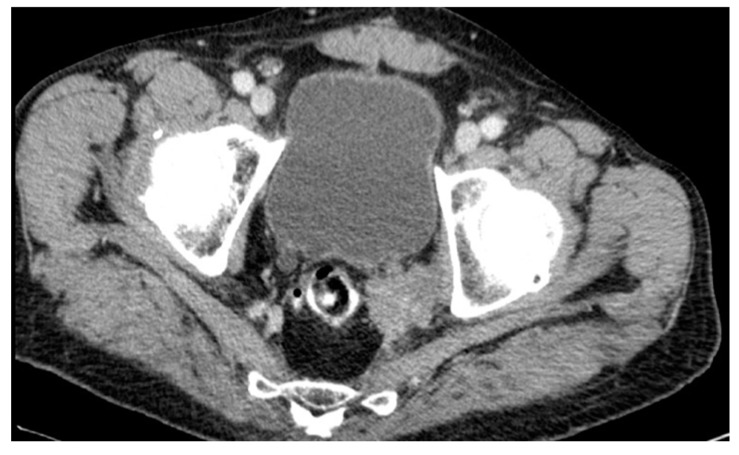
CT scan #1.

**Figure 8 cancers-16-00448-f008:**
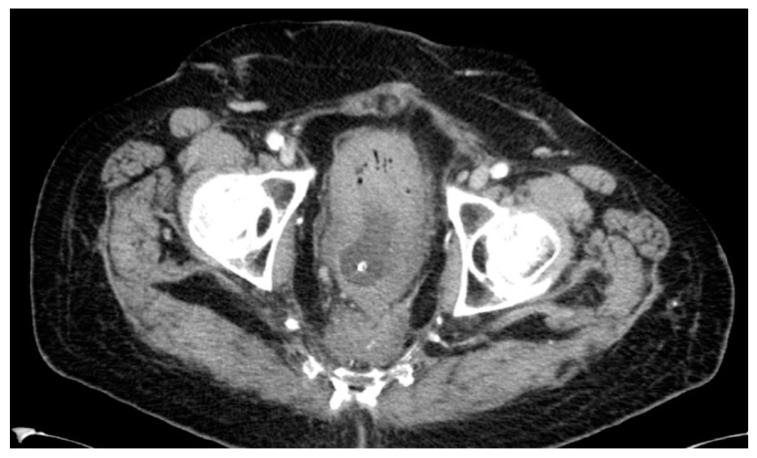
CT scan #2.

**Table 1 cancers-16-00448-t001:** Patient characteristics.

Patient Characteristics	
Number of Patients	188
Age (years, median)	63 (32–88)
Follow-up (months, median)	88.5
Distance from anal verge < 6 cm	75 (40%)
Sphincter preservation	149 (81%)
Sex	
Male	104 (55%)
Female	84 (45%)
Comorbidities	
Arterial hypertension	70 (37%)
Diabetes	19 (10%)
Obesity	21 (11%)
Inpatient stay (days, mean)	17.3 (7–80)
Complications	
Wound infection	29 (15%)
Anastomotic leak	27 (14%)
Stoma-associated complications	15 (8%)

**Table 2 cancers-16-00448-t002:** Pathological stage.

Pathological Stage	
ypT stage	
ypT0	32 (17%)
ypT1	17 (9%)
ypT2	56 (30%)
ypT3	77 (41%)
ypT4	6 (3%)
ypN stage	
ypN+	44 (24%)
ypN−	144 (77%)
UICC stage	
Stadium 0	30 (16%)
Stadium I	60 (32%)
Stadium IIa	49 (26%)
Stadium IIb	4 (2%)
Stadium IIIa	12 (6%)
Stadium IIIb	20 (11%)
Stadium IIIc	10 (5%)
Stadium IV	3 (2%)

**Table 3 cancers-16-00448-t003:** Recurrence.

Patients with Recurrence	
Total recurrences	38 (20.2%)
Patients with liver metastases	12 (6.4%)
Patients with lung metastases	7 (3.7%)
Patients with local recurrence	15 (8%)
Patients with multiple sites of recurrence	3 (1.6%)
Time to recurrence (months, median)	19
Distant metastases	17
Local recurrences	23
Local recurrences (cumulative)	15
<5 years of follow-up	11 (5.9%)
5–8 years of follow-up	11 (5.9%)
>8 years of follow-up	15 (8%)

**Table 4 cancers-16-00448-t004:** Local recurrence.

Age	Sex	Stage	Time to Recurrence	Location of Recurrence
50 years	Female	ypT3ypN2a	6 months	Lateral/nodal
68 years	Female	ypT2ypN0	12 months	Presacral
59 years	Female	ypT2ypN0	14 months	Presacral
77 years	Female	ypT4ypN0	15 months	Presacral
61 years	Male	ypT3ypN1	16 months	Presacral
75 years	Female	ypT3ypN2a	21 months	Anterior compartment
46 years	Male	ypT3ypN2	22 months	Presacral
49 years	Female	ypT3ypN2	23 months	Presacral
81 years	Male	ypT2ypN0	35 months	Presacral
75 years	Male	ypT3ypN0	39 months	Presacral
58 years	Male	ypT2ypN0	49 months	Presacral
73 years	Female	ypT2ypN0	99 months	Endoluminal
68 years	Female	ypT2ypN0	108 months	Anterior compartment
68 years	Male	ypT3ypN0	126 months	Presacral
76 years	Male	ypT2ypN1	135 months	Lateral/nodal

**Table 5 cancers-16-00448-t005:** Survival.

Survival				
	*n*	5-Year OS	10-Year OS	*p*
All patients	188	79.2%	58.5%	
Nodal Status				*p* = 0.001
ypN−	144 (76.6%)	84.1%	64.0%	
ypN+	44 (23.4%)	63.0%	40.9%	
Recurrence				*p* < 0.001
No recurrence	150 (79.8%)	89.6%	66.0%	
Recurrence	38 (20.2%)	41.3%	26.7%	
Location				*p* = 0.03
Upper third	8 (4.3%)	85.7%	58.3%	
Middle third	105 (55.9%)	83.8%	77.3%	
Lower Third	75 (39.9%)	72.1%	67.2%	
T stage				*p* = 0.063
ypT0	32 (17.0%)	86.4%	60.6%	
ypT1	17 (9.0%)	100%	80.8%	
ypT2	56 (29.8%)	79.2%	59.4%	
ypT3	77 (41.0%)	72.7%	50.8%	
ypT4	6 (3.2%)	60%		

## Data Availability

The data presented in this study are available on request from the corresponding author.

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
