# Peer review of "Late Local Recurrence after Neoadjuvant Therapy and Radical Resection for Locally Advanced Rectal Cancer"

_cancers, 2024, doi:10.3390/cancers16020448_

Round 1

Reviewer 1 Report (Previous Reviewer 1)

Comments and Suggestions for Authors

corrections approved

Author Response

No additional corrections necessary, corrections approved

Reviewer 2 Report (New Reviewer)

Comments and Suggestions for Authors

The authors conducted a study on 188 patients with rectal cancer who underwent RCT (radiochemotherapy) followed by radical resection. With an extensive follow-up period, they discovered that 38 patients experienced recurrent disease, with the following distribution: 8.0% local recurrence, 6.4% in the liver, and 3.7% in the lung. Notably, 68% of all recurrences occurred within the initial two years post-surgery. Intriguingly, four patients experienced recurrence more than nine years after their surgery, totaling 23 cases. These findings present a potential shift in perspective for the development of follow-up programs.

The manuscript contains many details that need to be revised, see below.

Major concerns:

1.      I think that neoadjuvant therapy including RT alone or CCRT or TNT policy is used for more tumor regression, improved local control and survival. The sentence this article use “completely prevent local recurrence “is not suitable use because no treatment guarantee to do it. Please revise it.

2.      I am really confused in table 3 about the event number of local recurrence between different follow up times. There are 15 patients with recurrent disease >9 years instead of 4 patients you mentioned before? Moreover, in table 4, only 2 patients had recurrent disease >9 years (108 months). Please clarify it.

3.      Why author excluded patients with R1/2 resection after chemoradiation? If authors want to discuss the efficiency of neoadjuvant treatment, all patients should be enrolled for a comprehensive survey.

4.      The major issue in this manuscript is late local recurrence. However, there is no risk or survival analysis about this? Did patients with ypT3-4 or N + or without adjuvant chemotherapy had a higher late local recurrence? Readers would be also interested in their patients with late local recurrence may had a lower survival or not? Please also provide more discussion and compared with previous literatures about late local recurrence. 

5. Finally, limited case numbers make the results convincing.

Author Response

1. Summary

Thank you very much for taking the time to review this manuscript. Please find the detailed responses below and the corresponding revisions/corrections in track changes in the re-submitted files.

2. Questions for General Evaluation

Reviewer’s Evaluation

Response and Revisions

Does the introduction provide sufficient background and include all relevant references?

Yes

Are all the cited references relevant to the research?

Yes

Is the research design appropriate?

Can be improved

Are the methods adequately described?

Must be improved

Are the results clearly presented?

Must be improved

Table3 rearranged

Are the conclusions supported by the results?

Must be improved

3. Point-by-point response to Comments and Suggestions for Authors

Comments 1: The authors conducted a study on 188 patients with rectal cancer who underwent RCT (radiochemotherapy) followed by radical resection. With an extensive follow-up period, they discovered that 38 patients experienced recurrent disease, with the following distribution: 8.0% local recurrence, 6.4% in the liver, and 3.7% in the lung. Notably, 68% of all recurrences occurred within the initial two years post-surgery. Intriguingly, four patients experienced recurrence more than nine years after their surgery, totaling 23 cases. These findings present a potential shift in perspective for the development of follow-up programs.

I think that neoadjuvant therapy including RT alone or CCRT or TNT policy is used for more tumor regression, improved local control and survival. The sentence this article use “completely prevent local recurrence “is not suitable use because no treatment guarantee to do it. Please revise it.

Response 1: Thank you very much for this suggestion. We know that this statement is somehow provocative, but it might demonstrates where this could go, that to say local control following neoadjuvant treatment is not as good as we expect it. However we revised this statement. (page1, paragraph1, line 17; page1, paragraph2, line25; page12, paragraph3, line305)

Comments 2: I am really confused in table 3 about the event number of local recurrence between different follow up times. There are 15 patients with recurrent disease >9 years instead of 4 patients you mentioned before? Moreover, in table 4, only 2 patients had recurrent disease >9 years (108 months). Please clarify it.

Response 2: Thank you for pointing this out. Table 3 shows the cummulative appearance of local recurrences, 11 local recurrences in the first 5 years of follow up, no additional recurrences between 5 and 8.3 years, and another 4 local recurrences beyond 8.3 years after surgery, which results in 15 local recurrences all together. We have rearranged table 3 to make this clearer.

The reviewer is right, the first so called “late local recurrence” was diagnosed 99 months after surgery, we have rounded that up to 9 years. To make this statement more accurate we have now rounded it down to 8 years, but that does not decrease the informative value of our study.

(page1, paragraph1, line16; page1, paragraph2, Line24; page5, table3; page5, paragraph2, line132)

Comments 3: Why author excluded patients with R1/2 resection after chemoradiation? If authors want to discuss the efficiency of neoadjuvant treatment, all patients should be enrolled for a comprehensive survey.

Response 3: We intended to study the potential role of neoadjuvant treatment in recurrent disease, especially in prolonging intervals to local failure. It was therefore a major goal to eliminate other factors potentially compromising our results such as suboptimal surgical resections. We considered R1/R2 resections as rather palliative than curative treatment and local relapse in theses patients as rather progressive than recurrent disease. It may well be that the overall recurrence rate is higher if R1 resection is considered “already recurred”, however we have found no evidence that this is common practice in rectal cancer RCTs.

Comments 4: The major issue in this manuscript is late local recurrence. However, there is no risk or survival analysis about this? Did patients with ypT3-4 or N + or without adjuvant chemotherapy had a higher late local recurrence? Readers would be also interested in their patients with late local recurrence may had a lower survival or not? Please also provide more discussion and compared with previous literatures about late local recurrence.

Response 4: According to the small number of patients with late local recurrences in this study, survival analyses would have been inappropriate and would not reveal significant differences. As we showed in table3, only one of the patients with late local recurrences initially had a ypT3 tumor and only one patient was node positive. None of the patients with late local recurrence received adjuvant treatment due to various reasons, we stated that in the results section. This could be a bias since only 50% of the whole study cohort received adjuvant treatment (page3, paragraph1, line79-82). Table5 depicts all significant differences concerning survival of the whole study cohort, as shown, the development of recurrence in general results in a poorer 5 and 10 year overall survival.

Comments 5: Finally, limited case numbers make the results convincing.

Response 5: The reviewer is right in pointing out that this case series is only what it is, a thoroughly documented retrospective series in which we try to summarize local experience with a focus on late recurrence.

Round 2

Reviewer 2 Report (New Reviewer)

Comments and Suggestions for Authors

The authors mad substantial efforts to implement the reviewers' suggestions and the manuscript has improved.

This manuscript is a resubmission of an earlier submission. The following is a list of the peer review reports and author responses from that submission.

Round 1

Reviewer 1 Report

Comments and Suggestions for Authors

statement on p 1 l. 32 regarding RCT enables SPS is not proven in most RCT.

in  modern TNT, the R0 resection rate was not proven, please correct your statement. p. 1 l. 34

DSF was improved in RAPIDO and Prodige 23 please correct your statement on p 1 l.40

It will be interesting to list the site of local recurrence in this population such as anastomotic, pre sacral, anterior compartiment, nodal in this 15 patients.

Reviewer 2 Report

Comments and Suggestions for Authors

Introduction

The authors observe that a small number of patients can develop local tumour recurrence more than 5 years after surgery, and suggest that such patients would benefit from longer surveillance. However, one is unable to predict which patients will suffer such a long-term recurrence, and thus recommendations should be generic for all patients.

The authors should stay focused on the purpose of their paper – local recurrences can occur 5 years or more after surgery and neo-adjuvant treatments. The introduction covers many areas that are confusing, or provides statements not supported by evidence. For example, Reference 7 – introduction – TNT protocols have shown improved rates of disease free survival, but possibly worse rates of local recurrence (as mentioned at the end of the discussion) and no overall survival benefit (e.g., RAPIDO) The authors should more accurately state the current narrative for neo-adjuvant therapy prior to rectal cancer surgery for radiation alone, chemoradiation, or TNT. The overall story is quite confusing and in some cases contradictory. In the discussion, it is intimated that late recurrence should be considered when clinicians make recommendations to patients ?? for TNT or even Non-operative Management (presumably for patients with a complete response). Again, it is not clear from the introduction what the paper will be concluding.

As well, an inaccurate statement suggests neo-adjuvant radiation can lower rates of permanent stoma - RCT’s comparing neo-adjuvant radiation before versus after surgery (Sauer et al trial – overall results, not select sub-study involving surgeon perceptions), or, radiation versus surgery alone (Dutch TME, MRC-CR07 (latter did use post RT for +ive margins), have not shown a lower rate of permanent stoma related to radiation use. This false common statement should be removed from the introduction.

Reference 5 refers to patients with recurrence or locally advanced (fixed) tumours, and is not relevant to this paper which includes only patients with non-metastatic R0 resection. The line involving this reference is not relevant to the paper.

This reviewer is unaware of evidence from RCT’s that second line chemotherapy can extend survival in patients with recurrent rectal cancer.

There are many statements in the introduction, repeated in the discussion, that are not supported by evidence.

Methods

The authors should describe why patients received various forms of neo-adjuvant therapy - ? stage II and III pre-op status based on CT ? MRI ? physical exam?

The authors should justify why they excluded R1R2 specimens. If readers are to use these findings to inform their own care, it is more helpful to include all patients receiving RCT for the years studied.

Results

The local recurrence rate of 8.7% is consistent with recurrence rates in other case series, but the exclusion of patients with R1 and R2 specimens suggests the overall rate of local recurrence in this case series is higher than 8.7%.

It would be helpful to have a table providing patient, tumour, and treatment variables for cases with and without local recurrence. Both types of nodal status (clinical (pre-radiation) and pathologic) could be presented. It may also be helpful to divide the data by year (2000-2009 vs 2010-2017). It is likely with the use of TME rates of recurrence will be different for the two time periods.

Overall

What then is the main purpose of this paper? It is recognized that neo-adjuvant therapies will delay presentation of local recurrence. The unique aspect of this case series is the presentation of the four ‘very late’ recurrences. The paper should focus on this. The inclusion of R1R2 patients in analyses would likely increase the uniqueness of such very late local recurrences.
